# ONLINE TESTING OF SUBGROUP TREATMENT EFFECTS BASED ON VALUE DIFFERENCE

## ABSTRACT

Online A/B testing plays a critical role in high-tech industry to guide product development and accelerate innovation. It performs a null hypothesis statistical test to determine which variant is better. However, a typical A/B test presents two problems: (i) a fixed-horizon framework inflates the false positive errors under continuous monitoring; (ii) the homogeneous effects assumption fails to identify a subgroup with a beneficial treatment effect. In this paper, we propose a sequential test for subgroup treatment effects based on value difference, named SUBTLE, to address these two problems simultaneously. The SUBTLE allows the experimenters to "peek" the results during the experiment without harming the statistical guarantees. It assumes heterogeneous treatment effects and aims to test if some subgroup of the population will benefit from the investigative treatment. If the testing result indicates the existence of such subgroup, a subgroup will be identified using a readily available estimated optimal treatment rule. We examine the empirical performance of our proposed test on both simulations and a real data set. The results show that the SUBTLE has high detection power with controlled type I error at any time, is more robust to noise covariates, and can achieve early stopping compared with the corresponding fixed-horizon test.

## 1 INTRODUCTION

Online A/B testing, as a kind of randomized control experiments, is widely used in high-tech industry to assess the value of ideas in a scientific manner (Kohavi et al., 2009). It randomly exposes users to one of the two variants: *control* (A), the currently-used version, or *treatment* (B), a new version being evaluated, and collects the metric of interest, such as conversion rate, revenue, etc. Then, a null hypothesis statistical test is performed to evaluate whether there is a statistically significant difference between the two variants on the metric of interest. This scientific design helps to control for the external variations and thus establish the causality between the variants and the outcome. However, the current A/B testing has its limitations in terms of framework and model assumptions.

First of all, most A/B tests employ a *fixed-horizon* framework, whose validity requires that the sample size should be fixed and determined before the experiment starts. However, experimenters, driven by a fast-paced product evolution in practice, often "peek" the experiment and hope to find the significance as quickly as possible to avoid large (i) time cost: an A/B test may take prohibitively long time to collect the determined size of samples; and (ii) opportunity cost: the users who have been assigned to a suboptimal variant will be stuck in a bad experience for a long time (Ju et al., 2019). The behaviors of continuously monitoring and concluding the experiment prematurely will be favorably biased towards getting significant results and lead to very high false positive probabilities, well in excess of the nominal significance level $\alpha$ (Goodson, 2014; Simmons et al., 2011). Another limitation of A/B tests is that they assume homogeneous treatment effects among the population and mainly focus on testing the average treatment effect. However, it is common that treatment effects vary across sub-populations. Testing the subgroup treatment effects will help decision makers distinguish the sub-population that may benefit from a particular treatment from those who may not, and thereby guide companies' marketing strategies in promoting new products.

The first problem can be addressed by applying the sequential testing framework. *Sequential testing*, contrast to the classic fixed-horizon test, is a statistical testing procedure that continuously checks for significance at every new sample and stops the test as soon as a significant result is detected,

while controlling the type I error at any time. It generally gives a significant decrease in the required sample size compared to the fixed-horizon test with the same type I error and type II error control, and thus is able to end an experiment much earlier. This field was first introduced by Wald (1945), who proposed *sequential probability ratio test* (SPRT) for simple hypotheses using *likelihood ratio* as the test statistics, and then was extended to composite hypotheses by many following literature (Schwarz, 1962; Armitage et al., 1969; Cox, 1963; Robbins, 1970; Lai, 1988). A thorough review is given in Lai (2001). However, the advantage of sequential testing in online A/B testing has not been recognized until recently Johari et al. (2015) brought the mSPRT, a variant of SPRT to A/B tests.

The second problem shows a demand for a test on subgroup treatment effects. Although sequential testing is rapidly developing in online A/B test, few work focuses on subgroup treatment effect testing. Yu et al. (2020) proposed a *sequential score test* (SST) based on score statistics under a *generalized linear model*, which aims to test if there is difference between treatment and control groups among any subjects. However, this test is based on a restrictive parametric assumption on treatment-covariates interaction and can't be used to test the subgroup treatment effects.

In this paper, we consider a flexible model, and propose a sequential test for *SUBgroup Treatment effects based on vaLuE difference* (SUBTLE), which aims to test if some group of the population would benefit from the investigative treatment. Our method does not require to specify any parametric form of covariate-specific treatment effects. If the null hypothesis is rejected, a beneficial subgroup can be easily obtained based on the estimated optimal treatment rule.

The remainder of this paper is structured as follows. In Section 2, we review the idea of the mSPRT and SST, and discuss how they are related to our test. Then in Section 3, we introduce our proposed method SUBTLE and provide the theoretical guarantee for its validity. We conduct simulations in Section 4 and real data experiments in Section 5 to demonstrate the validity, detection power, robustness and efficiency of our proposed test. Finally, in Section 6, we conclude the paper and present future directions.

## 2 RELATED WORK

### 2.1 MIXTURE SEQUENTIAL PROBABILITY RATIO TEST

The *mixture sequential probability ratio test* (mSPRT) (Robbins, 1970) supposes that the independent and identically distributed (i.i.d.) random variables $Y_1, Y_2, \cdots$ have a probability density function $f_\theta(x)$ induced by parameter $\theta$, and aims to test

$$H_0 : \theta = \theta_0 \quad v.s. \quad H_1 : \theta \neq \theta_0. \tag{1}$$

Its test statistics $\Lambda_n^\pi$ at sample size $n$ is a mixture of likelihood ratios as below:

$$\Lambda_n^\pi = \int_\Theta \prod_{i=1}^n \left( \frac{f_\theta(Y_i)}{f_{\theta_0}(Y_i)} \right) \pi(\theta) d\theta, \tag{2}$$

with a mixture density $\pi(\cdot)$ over the parameter space $\Theta$. The mSPRT stops the sampling at the stage

$$N = \inf\{n \geq 1 : \Lambda_n^\pi \geq 1/\alpha\} \tag{3}$$

and rejects the null hypothesis $H_0$ in favor of $H_1$. If no such time exists, it continues the sampling indefinitely and accept the $H_0$. Since the likelihood ratio under $H_0$ is a nonnegative martingale with initial value equal to 1, and so is the mixture of such likelihood ratios $\Lambda_n^\pi$, the type I error of mSPRT can be proved to be always controlled at $\alpha$ by an application of *Markov's inequality* and *optional stopping theorem*: $\mathbb{P}_{H_0}(\Lambda_n^\pi \geq \alpha^{-1}) \leq \frac{\mathbb{E}_{H_0}[\Lambda_n^\pi]}{\alpha^{-1}} = \frac{\mathbb{E}_{H_0}[\Lambda_0^\pi]}{\alpha^{-1}} = \alpha$. Besides, mSPRT is a test of power one (Robbins & Siegmund, 1974), which means that any small deviation from $\theta_0$ can be detected as long as waiting long enough. It is also shown that mSPRT is almost optimal for data from an exponential family of distributions, with respect to the expected stopping time (Pollak, 1978).

The mSPRT was brought to A/B test by Johari et al. (2015; 2017), who assume that the observations in control ($A = 0$) and treatment ($A = 1$) groups arrive in pairs $(Y_i^{(0)}, Y_i^{(1)})$, $i = 1, 2, \cdots$. They restricted their data model to the two most common cases in practice: normal distribution

and Bernoulli distribution, with $\mu_A$ and $\mu_B$ denoting the mean for control and treatment group, respectively. They test the hypothesis as below

$$H_0 : \theta := \mu_B - \mu_A = 0 \quad v.s. \quad H_1 : \theta \neq 0, \tag{4}$$

by directly applying mSPRT to the distribution of the differences $Z_i = Y_i^{(1)} - Y_i^{(0)}$ (normal), or the joint distribution of data pairs $(Y_i^{(0)}, Y_i^{(1)})$ (Bernoulli), $i = 1, 2, \cdots$. After making some approximations to the likelihood ratio and choosing a normal mixture density $\pi(\theta) \sim N(0, \tau^2)$, the test statistic $\Lambda_n^\pi$ is able to have a closed form for both normal and Bernoulli observations.

However, the mSPRT does not work well on testing heterogeneous treatment effects due to the complexity of likelihood induced by individual covariates. Specifically, a conjugate prior $\pi(\cdot)$ for the likelihood ratio may not exist anymore so that the computation for the test statistic is challenging. The unknown baseline covariates effect also increases the difficulty in constructing and approximating the likelihood ratios (Yu et al., 2020).

## 2.2 SEQUENTIAL SCORE TEST

The *sequential score test* (SST) (Yu et al., 2020) assumes a *generalized linear model* with a link function $g(\cdot)$ for the outcome $Y$

$$g(\mathbb{E}[Y|A, \mathbf{X}]) = \boldsymbol{\mu}^T \mathbf{X} + (\boldsymbol{\theta}^T \mathbf{X})A, \tag{5}$$

where $A$ and $\mathbf{X}$ denote the binary treatment indicator and user covariates vector, and tests the multi-dimensional treatment-covariates interaction effect:

$$H_0 : \boldsymbol{\theta} = \mathbf{0} \quad vs. \quad H_1 : \boldsymbol{\theta} \neq \mathbf{0}, \tag{6}$$

while accounting for the linear baseline covariates effect $\boldsymbol{\mu}^T \mathbf{X}$. For the test statistics $\Lambda_n^\pi$, instead of using a mixture likelihood ratios as mSPRT, SST employed a mixture asymptotic probability ratios of a score statistics. Since the probability ratio has the same martingale structure as the likelihood ratio, the type I error can still be controlled with the same decision rule as mSPRT (3). The asymptotic normality of the score statistics also guarantees a closed form of $\Lambda_n^\pi$ with a multivariate normal mixture density $\pi(\cdot)$. However, the considered parametric model (5) can only be used to test if there are linear covariate-treatment interaction effects, and may fail to detect the existence of a subgroup with enhanced treatment effects. In addition, the subgroup estimated based on the index $\theta^T \mathbf{X}_i$ may be biased if the assumed linear model (5) is misspecified. Therefore in this paper, we propose a subgroup treatment effect test, which is able to test the existence of a beneficial subgroup and does not require to specify the form of treatment effects.

## 3 SUBGROUP TREATMENT EFFECTS TEST BASED ON VALUE DIFFERENCE

### 3.1 PROBLEM SETUP

Suppose we have i.i.d. data $\mathbf{O}_i = \{Y_i, A_i, \mathbf{X}_i\}$, $i = 1, 2, \cdots$, where $Y_i, A_i, \mathbf{X}_i$ respectively denote the observed outcome, binary treatment indicator, and $p$-dimensional user covariates vector. Here, we consider a flexible generalized linear model:

$$g(\mathbb{E}[Y_i|A_i, \mathbf{X}_i]) = \mu(\mathbf{X}_i) + \theta(\mathbf{X}_i)A_i, \tag{7}$$

where baseline covariates effect $\mu(\cdot)$ and treatment-covariates interaction effect $\theta(\cdot)$ are completely unspecified functions, and $g(\cdot)$ is a prespecified link function. For example, we use the identity link $g(\mu) = \mu$ for normal response and the logit link $g(\mu) = \log \frac{\mu}{1-\mu}$ for binary data.

Assuming $Y$ is coded such that larger values indicate a better outcome, we consider the following test of subgroup treatment effects:

$$H_0 : \forall \mathbf{x} \in \mathcal{X}, \; \theta(\mathbf{x}) \leq 0 \; vs. \; H_1 : \exists \mathcal{X}_0 \subset \mathcal{X} \text{ such that } \theta(\mathbf{x}) > 0 \text{ for all } \mathbf{x} \in \mathcal{X}_0, \tag{8}$$

where $\mathcal{X}_0$ is the beneficial subgroup with $\mathbb{P}(\mathbf{X} \in \mathcal{X}_0) > 0$. Note that the above subgroup test is very different from the covariate-treatment interaction test considered in (6) and is much more challenging due to several aspects. First, both $\mu(\cdot)$ and $\theta(\cdot)$ are nonparametric and need to be estimated.

Second, the considered hypotheses are moment inequalities which are nonstandard. Third, it allows the nonregular setting, i.e. $\mathbb{P}\{\theta(\mathbf{X}) = 0\} > 0$, which makes associated inference difficult. Here, we propose a test based on value difference between the optimal treatment rule and a fixed treatment rule.

Let $V(d) = \mathbb{E}_{(Y^*(a), \mathbf{X})}\{Y^*(d(\mathbf{X}))\}$ denote a value function for a treatment decision rule, where $Y^*(d(\mathbf{X}))$ is the potential outcome if treatment were allocated according to the fixed treatment decision rule $d(\mathbf{X})$, which maps the information in $\mathbf{X}$ to treatment $\{0, 1\}$. Consider the value difference $\Delta = V(d^{\mathrm{opt}}) - V(0)$ between the optimal treatment rule $d^{\mathrm{opt}} = 1\{\theta(\mathbf{X}) > 0\}$ and the treatment rule that assigns control to everyone $d = 0$, where $1\{\cdot\}$ is an indicator function. If the null hypothesis is true, no one would benefit from the treatment and the optimal treatment rule assigns everyone to control, and therefore the value difference is zero. However, if the alternative hypothesis is true, some people would have higher outcomes being assigned to treatment and thus the value difference is positive. In this way the testing hypotheses (8) can be equivalently transformed into the following pair:

$$H_0 : \Delta = 0 \quad vs. \quad H_1 : \Delta > 0. \tag{9}$$

We make the following standard causal inference assumptions: (i) consistency, which states that the observed outcome is equal to the potential outcome under the actual treatment received, i.e. $Y = Y^*(1)I(A = 1) + Y^*(0)I(A = 0)$; (ii) no unmeasured confounders, i.e. $Y^*(a) \perp\!\!\!\perp A | \mathbf{X}$, which means the potential outcome is independent of treatment given covariates; (iii) positivity, i.e. $\mathbb{P}(A = a | \mathbf{X} = \mathbf{x}) > 0$ for $a = 0, 1$ and all $\mathbf{x} \in \mathcal{X}$ such that $\mathbb{P}(\mathbf{X} = \mathbf{x}) > 0$. Under these assumptions, it can be shown that

$$V(d) = \mathbb{E}_{\mathbf{X}}\{\mathbb{E}[Y | A = d(\mathbf{X}), \mathbf{X}]\}.$$

## 3.2 Algorithm and implementation

We take the *augmented inverse probability weighted* (AIPW) estimator (Robins et al., 1994; Zhang et al., 2012) for the value function of a given treatment rule $d$:

$$\hat{V}_{\mathrm{AIPW}}(d) = \frac{1}{n}\sum_{i=1}^{n}\left\{\frac{Y_i \cdot 1\{A_i = d\}}{p_{A_i}(\mathbf{X}_i)} - \left(\frac{1\{A_i = d\}}{p_{A_i}(\mathbf{X}_i)} - 1\right)\cdot \mathbb{E}[Y_i | A_i = d, \mathbf{X}_i]\right\}$$

where $p_A(\mathbf{X}) = A * p(\mathbf{X}) + (1 - A) * (1 - p(\mathbf{X}))$ and $p(\mathbf{X}) = \mathbb{P}(A = 1 | \mathbf{X})$ is the propensity score. This estimator is unbiased, i.e., $\mathbb{E}_{(Y, A, \mathbf{X})}[\hat{V}_{\mathrm{AIPW}}(d)] = V(d)$. Moreover, the most important property of AIPW estimator is the double robustness, that is, the estimator remains consistent if either the estimator of $\mathbb{E}[Y | A = d, \mathbf{X}]$ or the estimator of the propensity score $p(\mathbf{X})$ is consistent, which gives much flexibility. Then the value difference $\Delta$ is unbiased estimated by

$$D(\mathbf{O}_i; \mu, \theta, p) := \left\{\frac{1\{A_i = 1(\theta(\mathbf{X}_i) > 0)\}}{p_{A_i}(\mathbf{X}_i)} * Y_i - \left(\frac{1\{A_i = 1(\theta(\mathbf{X}_i) > 0)\}}{p_{A_i}(\mathbf{X}_i)} - 1\right)\right.$$
$$\left. * g^{-1}\left(\mu(\mathbf{X}_i) + \theta(\mathbf{X}_i)1(\theta(\mathbf{X}_i) > 0)\right)\right\} - \left\{\frac{1(A_i = 0)}{1 - p(\mathbf{X}_i)} * Y_i - \left(\frac{1(A_i = 0)}{1 - p(\mathbf{X}_i)} - 1\right) * g^{-1}\left(\mu(\mathbf{X}_i)\right)\right\} \tag{10}$$

where $g^{-1}(\cdot)$ is the inverse of the link function. That is,

$$\mathbb{E}_{(Y, A, \mathbf{X})}[D(\mathbf{O}_i; \mu, \theta, p)] = \Delta.$$

Since $\mu(\cdot)$, $\theta(\cdot)$ and $p(\cdot)$ are usually unknown, we let data come in batches and estimate them based on previous batches of data. Algorithm 1 shows our complete testing procedures.

In step (ii) of Algorithm 1, we estimate $\mu(\cdot)$ and $\theta(\cdot)$ by respectively building a random forest on control observations and on treatment observations in previous batches. The propensity score $p(\cdot)$ is estimated by computing the proportion of treatment observations ($A = 1$) in previous batches. In step (iv) we estimate $\sigma_k$ with $\hat{\sigma}_k = \sqrt{\frac{s_k^2}{m}}$, where $s_k^2$ is the sample variance of $D(\mathbf{O}_i; \hat{\mu}_{k-1}, \hat{\theta}_{k-1}, \hat{p}_{k-1})$, $\forall \mathbf{O}_i \in \overline{C}_{k-1}$. Note that $R_k$ (14) is a multiplier of an asymptotic unbiased estimator for $\Delta$, which is defined as below:

$$\hat{\Delta}_k := \frac{\sum_{j=1}^{k} \hat{\sigma}_j^{-1} \bar{D}_j(C_j; \overline{C}_{j-1})}{\sum_{j=1}^{k} \hat{\sigma}_j^{-1}}. \tag{11}$$

---

**Algorithm 1:** Subgroup treatment effects sequential test based on value difference

---

1. Initialize $k = 0$, $\Lambda_k^\pi = 0$. Choose a significance level $0 < \alpha < 1$, a batch size $m$, an initial batch size $l$, and a failure time $M$.
2. Sample $l$ observations to formulate initial batch $\mathcal{C}_0$.

**while** *True* **do**

    (i) k=k+1;

    (ii) Let $\overline{\mathcal{C}}_{k-1} = \cup_{j=0}^{k-1} \mathcal{C}_j$. Estimate $\mu(\cdot)$, $\theta(\cdot)$ and $p(\cdot)$ based on data in $\overline{\mathcal{C}}_{k-1}$ to get $\hat{\mu}_{k-1}$, $\hat{\theta}_{k-1}$ and $\hat{p}_{k-1}$;

    (iii) Sample another $m$ observations to formulate batch $\mathcal{C}_k$. For each $\mathbf{O}_i \in \mathcal{C}_k$, calculate $D(\mathbf{O}_i; \hat{\mu}_{k-1}, \hat{\theta}_{k-1}, \hat{p}_{k-1})$. Let

$$\bar{D}_k(\mathcal{C}_k; \overline{\mathcal{C}}_{k-1}) = \frac{1}{m} \sum_{\mathbf{O}_i \in \mathcal{C}_k} D(\mathbf{O}_i; \hat{\mu}_{k-1}, \hat{\theta}_{k-1}, \hat{p}_{k-1}) \quad (13)$$

    (iv) Estimate the conditional standard deviation $\sigma_k = sd\left(\bar{D}_k(\mathcal{C}_k; \overline{\mathcal{C}}_{k-1}) | \overline{\mathcal{C}}_{k-1}\right)$ based on data in $\overline{\mathcal{C}}_{k-1}$ and denote it as $\hat{\sigma}_k$;

    (v) Calculate

$$R_k = \frac{1}{\sqrt{k}} \sum_{j=1}^k \hat{\sigma}_j^{-1} \bar{D}_j(\mathcal{C}_j; \overline{\mathcal{C}}_{j-1}) \quad (14)$$

    and

$$\Lambda_k^\pi = \int \frac{\psi_{\left(\frac{1}{\sqrt{k}}(\sum_{j=1}^k \hat{\sigma}_j^{-1})\Delta, \, 1\right)}(R_k)}{\psi_{(0,\,1)}(R_k)} \pi(\Delta) d\Delta, \quad (15)$$

    where $\psi_{(\mu, \sigma^2)}(\cdot)$ denotes the probability density function of a normal distribution with mean $\mu$ and variance $\sigma^2$;

    **if** $\Lambda_k^\pi > 1/\alpha$ *or* $k \times m + l > M$ **then**

        | break;

    **end**

**end**

**if** $\Lambda_k^\pi > 1/\alpha$ **then**

    Reject $H_0$. Estimate $\theta(\cdot)$ using all the data up to now and identify a subgroup $1\{\hat{\theta}(\mathbf{X}) > 0\}$;

**else**

    | Accept $H_0$;

**end**

---

In section 3.3 we will show that $R_k$ has an asymptotic normal distribution with same variance but different means under null and local alternatives, so that our test statistics $\Lambda_k^\pi$ (15) is a mixture asymptotic probability ratios of $R_k$. Since the value difference is always non-negative, we choose a truncated normal $\pi(\Delta) = \frac{2}{\sqrt{2\pi\tau^2}} \cdot \exp\left\{-\frac{\Delta^2}{2\tau^2}\right\} \cdot 1(\Delta > 0)$ as the mixture density, where $\tau^2$ is estimated based on historical data. The simulation result in Appendix A.2.1 shows considerable robustness in choosing $\tau^2$. Our test statistic now has a closed form:

$$\Lambda_k^\pi = 2 \left\{ \frac{k}{k + (\tau \cdot \sum_{j=1}^k \hat{\sigma}_j^{-1})^2} \right\}^{1/2} \times \exp\left\{ \frac{(\tau \cdot \sum_{j=1}^k \hat{\sigma}_j^{-1} \cdot R_k)^2}{2\left[(\tau \cdot \sum_{j=1}^k \hat{\sigma}_j^{-1})^2 + k\right]} \right\} \times [1 - F(0)], \quad (12)$$

where $F(\cdot)$ is the cumulative distribution function of a normal distribution with mean $\frac{\sqrt{k} \cdot \sum_{j=1}^k \hat{\sigma}_j^{-1} R_k}{(\sum_{j=1}^k \hat{\sigma}_j^{-1})^2 + k}$ and variance $\frac{k\tau^2}{\tau^2(\sum_{j=1}^k \hat{\sigma}_j^{-1})^2 + k}$.

If the null hypothesis is rejected, we can employ random forests to estimate $\theta(\cdot)$ based on all the data up to the time that the experiment ends. Then the optimal treatment rule $\hat{\theta}(\mathbf{x})$ naturally gives the beneficial subgroup $\mathcal{X}_0 = \{\mathbf{x} : \hat{\theta}(\mathbf{x}) > 0\}$.

## 3.3 VALIDITY

In this section, we will show that our proposed test SUBTLE is able to control type I error at any time, that is, $\mathbb{P}_{H_0}(\Lambda_k^\pi > 1/\alpha) < \alpha$ for any $k \in \mathbb{N}$. As we discussed in Section 2.1, if we can show that the ratio term in $\Lambda_k^\pi$ (15) has a martingale structure under $H_0$, it follows easily that the type I error is always controlled at $\alpha$. Theorem 3.1 gives the respective asymptotic distributions of $R_k$ under null and local alternative, which demonstrates that the test statistics $\Lambda_k^\pi$ is a mixture asymptotic probability ratios weighted by $\pi(\cdot)$. Proposition 1 shows that this asymptotic probability ratio is a martingale when the sample size is large enough. Combining these two results with the demonstration in Section 2.1, we can conclude that the type I error of SUBTLE is always controlled at $\alpha$.

We assume the following conditions hold:

- (C1) $k$ diverges to infinity as sample size $n$ diverges to infinity.
- (C2) Lindeburg-like condition:

$$\frac{1}{k}\sum_{j=1}^{k}\mathbb{E}\left[\left(\frac{\bar{D}_j(\mathcal{C}_j;\overline{\mathcal{C}}_{j-1})}{\hat{\sigma}_j}\right)^2 \cdot \mathbf{1}\left(\frac{|\bar{D}_j(\mathcal{C}_j;\overline{\mathcal{C}}_{j-1})|}{\hat{\sigma}_j} > \epsilon\sqrt{k}\right)\bigg|\overline{\mathcal{C}}_{j-1}\right] = o_p(1)$$

  for all $\epsilon > 0$.

- (C3) $\frac{1}{k}\sum_{i=1}^{k}\frac{\sigma_j^2}{\hat{\sigma}_j^2} \xrightarrow{p} 1$.

- (C4) $\quad \frac{1}{k}\sum_{j=1}^{k}\hat{\sigma}_j^{-1}\left(\mathbb{E}[\bar{D}_j(\mathcal{C}_j;\overline{\mathcal{C}}_{j-1})|\overline{\mathcal{C}}_{j-1}] - \mathbb{E}[\bar{D}_j(\mathcal{C}_j;\hat{d}_{j-1}^{opt},\mu,\theta,p)|\overline{\mathcal{C}}_{j-1}]\right) \quad = o_p(k^{-1/2})$.

- (C5) $\frac{1}{k}\sum_{j=1}^{k}\hat{\sigma}_j^{-1}\left(\mathbb{E}[\bar{D}_j(\mathcal{C}_j;\hat{d}_{j-1}^{opt},\mu,\theta,p)|\overline{\mathcal{C}}_{j-1}] - \Delta\right) = o_p(k^{-1/2})$.

**Theorem 3.1** *For $\hat{\Delta}_k$ defined in (11), under conditions (C1)-(C5),*

$$\frac{1}{\sqrt{k}}\left(\sum_{j=1}^{k}\hat{\sigma}_j^{-1}\right)\left(\hat{\Delta}_k - \Delta\right) \xrightarrow{d} N(0,1) \quad \text{as } k \to \infty, \tag{16}$$

*where $\xrightarrow{d}$ represents convergence in distribution. In particular, as $k \to \infty$, $R_k \xrightarrow[H_0]{d} N(0,1)$ under null hypothesis $\Delta = 0$, while $R_k - \frac{1}{\sqrt{k}}\left(\sum_{j=1}^{k}\hat{\sigma}_j^{-1}\right)\Delta \xrightarrow[H_1]{d} N(0,1)$ under local alternative $\Delta = \frac{\delta}{\sqrt{k}}$, where $\delta > 0$ is fixed.*

**Proposition 1** *Let $\lambda_k = \frac{\psi_{\left(\frac{1}{\sqrt{k}}\left(\Sigma_{j=1}^{k}\hat{\sigma}_j^{-1}\right)\Delta,\,1\right)}(R_k)}{\psi_{(0,\,1)}(R_k)}$, and $\mathcal{F}_k$ denote a filtration that contains all the historical information in the first $(k+1)$ batches $\overline{\mathcal{C}}_k$. Then under null hypothesis $H_0 : \Delta = 0$, $\mathbb{E}[\lambda_{k+1}|\mathcal{F}_k]$ is approximately equal to $\lambda_k \cdot \exp\{o_p(1)\}$.*

The proofs of above results are given in the Appendix A.1.

## 4 SIMULATED EXPERIMENTS

In this section, we evaluate the test SUBTLE on three metrics: type I error, power and sample size. We first compare SUBTLE with SST in terms of type I error and power under five models in Section 4.1. Then in Section 4.2, we present the impact of noise covariates on their powers. Finally in Section 4.3, we compare the stopping time of SUBTLE to the required sample size of a fixed-horizon value difference test. The significance level $\alpha = 0.05$, initial batch size $l = 300$, failure time $M = 2300$ and variance of mixture distribution $\tau^2 = 1$ are fixed for all simulation settings.

## 4.1 TYPE I ERROR & POWER

We consider five data generation models in the form of (7) with logistic link $g(\cdot)$. Data are generated in batches with batch size $m = 20$ and are randomly assigned to two groups with fixed propensity score $p(\mathbf{X}) = 0.5$. Each experiment is repeated 1000 times to estimate the type I error and power. For the first four models, we choose

- Five covariates: $X_1 \overset{iid}{\sim} Ber(0.5)$, $X_2 \overset{iid}{\sim} Unif[-1, 1]$, $X_3, X_4, X_5 \overset{iid}{\sim} N(0, 1)$
- Two baseline effect: $\mu_1(\mathbf{X}) = -2 - X_1 + X_3^2$, $\mu_2(\mathbf{X}) = -1.3 + X_1 + 0.5X_2 - X_3^2$
- Two treatment-covariates interaction effect: $\theta_1(\mathbf{X}) = c \cdot 1\{X_1 + 2X_3 > 0\}$, $\theta_2(\mathbf{X}) = c \cdot 1\{X_2 > 0 \text{ or } X_5 < -0.5\}$.

Table 1: The first four models

| Model | Input covariates | $\mu(\mathbf{X})$ | $\theta(\mathbf{X})$ |
|---|---|---|---|
| I | $X_1, X_3$ | $\mu_1(\mathbf{X})$ | $\theta_1(\mathbf{X})$ |
| II | $X_1, X_2, X_3, X_4, X_5$ | $\mu_2(\mathbf{X})$ | $\theta_2(\mathbf{X})$ |
| III | $X_1, X_2, X_3, X_4, X_5$ | $\mu_1(\mathbf{X})$ | $\theta_2(\mathbf{X})$ |
| IV | $X_1, X_2, X_3, X_4, X_5$ | $\mu_2(\mathbf{X})$ | $\theta_1(\mathbf{X})$ |

Table 1 displays which covariates, $\mu(\mathbf{X})$ and $\theta(\mathbf{X})$ are employed in each model. For model V, we consider the following high-dimensional setting:

$X_r \overset{iid}{\sim} N(0.2r - 0.6, 1)$, $r = 1, 2, 3, 4, 5$ $\qquad X_{14} \overset{iid}{\sim} Unif[-0.5, 1.5]$

$X_r \overset{iid}{\sim} N(0.2r - 1.6, 2)$, $r = 6, 7, 8, 9, 10$ $\qquad X_{15} \overset{iid}{\sim} Unif[-1.5, 0.5]$

$X_r \overset{iid}{\sim} Unif[-0.5r + 5, 0.5r - 5]$, $r = 11, 12, 13$ $\quad X_r \overset{iid}{\sim} Ber(0.2r - 3.1)$, $r = 16, 17, 18, 19, 20$

$\mu(\mathbf{X}) = -0.8 + X_{18} + 0.5X_{12} - X_3^2$ $\qquad \theta(\mathbf{X}) = c \cdot 1\{(X_{14} > -0.1) \& (X_{20} = 1)\}$,

where $c$ varies among $\{-1, 0, 0.6, 0.8, 1\}$ indicating the intensity of the value difference. When $c = -1$ and $0$, the null hypothesis is true and the type I error is estimated, while when $c = 0.6, 0.8, 1$, the alternative is true and the power is estimated.

Table 2 shows that the SUBTLE is able to control type I error and achieve competing detection power, especially under high-dimensional setting (Model V); however, SST couldn't control type I error especially when $c = -1$. This can be explained by two things: (i) the linearity of model (5) is violated; (ii) SST is testing if there is difference between treatment and control groups among any subjects, instead of the existence of a beneficial subgroup. Specifically, SST is testing if the least false parameter $\boldsymbol{\theta}^*$, to which the MLE of $\boldsymbol{\theta}$ under model misspecification converges, is zero or not. We also perform experiments with batch size $m = 40$, and the results (shown in Appendix A.2.1) do not have much difference.

Table 2: Estimated type I error or power for SUBTLE and SST with batch size 20

| Model | I | | II | | III | | IV | | V | |
|---|---|---|---|---|---|---|---|---|---|---|
| $c$ | SUBTLE | SST | SUBTLE | SST | SUBTLE | SST | SUBTLE | SST | SUBTLE | SST |
| -1 | 0.009 | 0.695 | 0.002 | 0.589 | 0.003 | 0.224 | 0.004 | 0.411 | 0.002 | 0.008 |
| 0 | 0.015 | 0.134 | 0.010 | 0.023 | 0.006 | 0.095 | 0.010 | 0.023 | 0.006 | 0.038 |
| 0.6 | 0.323 | 0.564 | 0.491 | 0.513 | 0.269 | 0.389 | 0.424 | 0.425 | 0.559 | 0.170 |
| 0.8 | 0.623 | 0.845 | 0.878 | 0.900 | 0.719 | 0.723 | 0.822 | 0.824 | 0.925 | 0.390 |
| 1 | 0.911 | 0.974 | 0.988 | 0.996 | 0.952 | 0.943 | 0.985 | 0.982 | 0.997 | 0.742 |

## 4.2 NOISE COVARIATES

It is common in practice that a large number of covariates are incorporated in the experiment whereas the actual outcome only depends on a few of them. Some covariates do not have any effect on

the response, like $X_4$ in Model II, III, IV, and we call them noise covariates. In the following simulation, we explore the impact of noise covariates to the detection power. We choose Model I with $c = 0.8$ as the base model, and at each time add three noise covariates which are respectively from normal $N(0, 1)$, uniform $Unif[-1, 1]$, and Bernoulli $Ber(0.5)$ distributions. The batch size is set to $m = 40$ for computation efficiency. Figure 1 shows that SST has continuously decreasing powers as the number of noise covariates increases, while the power of SUBTLE is more robust to the noise covariates

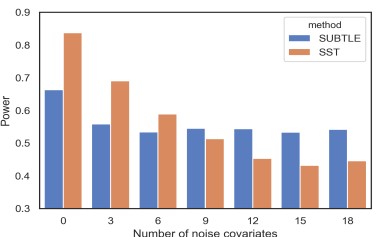

Figure 1: Estimated power v.s. the number of noise covariates

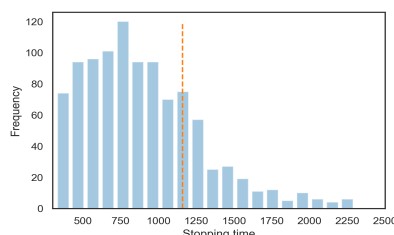

Figure 2: Histogram of stopping time for 1000 replicates of experiments under Model V with c=1

### 4.3 STOPPING TIME

A key feature for sequential test is that it has an expected smaller sample size than fixed-horizon test. For comparison, we consider a fixed-horizon version of SUBTLE, which leverages the Theorem 3.1 and rejects the null hypothesis $H_0 : \Delta = 0$ when $R_k > Z_\alpha$ for some predetermined $k$, where $Z_\alpha$ denotes the $(1 - \alpha)$ quantile of standard normal distribution. We assume $\sigma^{-1} = \lim_{k \to \infty} \frac{1}{k} \sum_{j=1}^{k} \hat{\sigma}_j^{-1}$, then the required number of batches $k$ can be calculated as $k = \frac{\sigma^2(Z_\alpha + Z_{1-\text{power}})^2}{\Delta^2}$, and thus the required sample size is $n = k * m + l$. The true value difference $\Delta$ can be directly estimated from data generated under true model and two treatment rules, while $\sigma^2$ is estimated by the sample variance of $\hat{\Delta}_{k'}$ (11) times $k'$ for some fixed large $k'$. Here, we choose Model V with $c = 1$ and batch size $m = 20$. The stopping sample size of our sequential SUBTLE over 1000 replicates are shown in Figure 2, and the dashed vertical line indicates the required sample size for the fixed-horizon SUBTLE with the same power 0.997 (seen from Table 2) under the same setting. We can find that most of the time our sequential SUBTLE arrives the decision early than the fixed-horizon version, but occasionally it can take longer. The distribution of the stopping time for sequential SUBTLE is right-skewed, which is line with the findings in Johari et al. (2015) and Ju et al. (2019).

## 5 REAL DATA EXPERIMENTS

We use Yahoo real data to examine the performance of our SUBTLE, which contains user click events on articles over 10 days. Each event has a timestamps, a unique article id (variants), a binary click indicator (response), and four independent user features (covariates). We choose two articles (id=109520 and 109510) with the highest click through rates as control and treatment, respectively. We set the significance level $\alpha = 0.05$, initial batch size and batch size $l = m = 200$, and the failure time $M = 50000$.

To demonstrate the false positive control of our method, we conduct A/A test and permutation test. For A/A test, we only use data on article 109510 and randomly generate fake treatment indicator. Our method accepts the null hypothesis. For permutation test, we use combined data from article 109510 and 109520, and permute their response 1000 times while leaving treatment indicator and covariates unchanged. The estimated false positive rate is below the significance level.

Then we test if there is any subgroup of users who would have higher click rate on article 109510. In this experiment, SUBTLE rejects the null hypothesis with sample size $n = 12400$. We identify the beneficial subgroup $1\{\theta(\mathbf{X}) > 0\}$ by estimating $\hat{\theta}(\mathbf{X})$ with random forest on the first 12400

observations. To get a structured optimal treatment rule, we then build a classification tree on the same 12400 samples with random forest estimator $1\{\hat{\theta}(\mathbf{X}) > 0\}$ as true labels. The resulting decision tree (shown in Appendix A.2.2) suggests that the users in the subgroup defined by $\{X_3 < 0.7094$ or $(X_3 \geq 0.7094, X_1 \geq 0.0318$ and $X_4 < 0.0003)\}$ benefit from treatment.

We then use the 50000 samples after the first 12400 samples as test data set, and then compute the difference of click through rates between article 109510 and 109520 on the test data (overall treatment effect), and the same difference in the subgroup of the test data (subgroup treatment effect). We found that the subgroup treatment effect 0.009 is larger than the overall treatment effect 0.006, which shows that the identified subgroup has enhanced treatment effects than the overall population.

We further compute the *inverse probability weighted* (IPW) estimator $\frac{1}{n}\sum_{i=1}^{n}\frac{1(A_i=d(\mathbf{X}_i))*Y_i}{p_{A_i}(\mathbf{X}_i)}$ using the test data for the values of two treatment rules: $d_1(\mathbf{X}) = 0$ that assigns everyone to control and the optimal treatment rule $d_2(\mathbf{X}) = 1\{\hat{\theta}(\mathbf{X}) > 0\}$ estimated by random forest. Their IPW estimates are respectively 0.043 and 0.049, which suggests that the estimated optimal treatment rule is better than the fixed rule that assigns all users to the control group. This implies there exists a subgroup of the population that does benefit from the article 109510.

## 6 CONCLUSION

In this paper, we propose SUBTLE, which is able to sequentially test if some subgroup of the population will benefit from the investigative treatment. If the null hypothesis is rejected, a beneficial subgroup can be easily identified based on the estimated optimal treatment rule. The validity of the test has been proved by both theoretical and simulation results. The experiments also show that SUBTLE has high detection power especially under high-dimensional setting, is robust to noise covariates, and allows quick inference most of time compared with fixed-horizon test.

Same as mSPRT and SST, the rejection condition of SUBTLE may never be reached under some cases, especially when the true effect size is negligible. Thus, a failure time is needed to terminate the test externally and accept the null hypothesis if we ever reach it. How to choose a failure time to trade off between waiting time and power need to be studied in the future. Another future direction is the application of our test under adaptive allocation, where users will have higher probabilities of being assigned to a beneficial variant based on previous observations. However, the validity may not be guaranteed anymore under adaptive allocation and more theoretical investigations are needed.

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

## A    APPENDIX

### A.1    PROOFS

#### A.1.1    PROOF OF THEOREM 3.1

Among the conditions for Theorem 3.1, (C1) holds by nature. We suppose that (C2) and (C3) hold. (C4) and (C5) depends on the convergence rate of estimators of $\mu, \theta, p$. Wager & Athey (2018) showed that under certain constraints on the subsampling rate, random forest predictions converge at the rate $n^{s-1/2}$, where $s$ is chosen to satisfy some conditions. We assume that under this rate, (C4) and (C5) also hold.

Let $\mathcal{F}_j, 0 \leq j \leq k$, denote a filtration generated by observations in first $(j+1)$ batches $\overline{\mathcal{C}}_j = \cup_{r=0}^{j} \mathcal{C}_r$, and $\bar{D}_j(\mathcal{C}_j; \hat{d}_{j-1}^{opt}, \mu, \theta, p)$ denote an AIPW estimator for $\Delta$ with only optimal decision rule estimated by previous batches:

$$
\bar{D}_j(\mathcal{C}_j; \hat{d}_{j-1}^{opt}, \mu, \theta, p) := \frac{1}{m} \sum_{\mathbf{O}_i \in \mathcal{C}_j} \left\{ \frac{1\left\{A_i = 1(\hat{\theta}_{j-1}(\mathbf{X}_i) > 0)\right\}}{p_{A_i}(\mathbf{X}_i)} * Y_i - \left( \frac{1\left\{A_i = 1(\hat{\theta}_{j-1}(\mathbf{X}_i) > 0)\right\}}{p_{A_i}(\mathbf{X}_i)} - 1 \right) \right.
$$
$$
\left. * g^{-1}\left( \mu(\mathbf{X}_i) + \theta(\mathbf{X}_i)1(\hat{\theta}_{j-1}(\mathbf{X}_i) > 0) \right) \right\}
$$
$$
- \left\{ \frac{1(A_i = 0)}{1 - p(\mathbf{X}_i)} * Y_i - \left( \frac{1(A_i = 0)}{1 - p(\mathbf{X}_i)} - 1 \right) * g^{-1}\left( \mu(\mathbf{X}_i) \right) \right\}. \tag{17}
$$

Then

$$\frac{1}{k}\left(\sum_{j=1}^{k}\hat{\sigma}_j^{-1}\right)\left(\hat{\Delta}_k - \Delta\right) \tag{18}$$

$$= \frac{1}{k}\sum_{j=1}^{k}\hat{\sigma}_j^{-1}\left(\bar{D}_j(\mathcal{C}_j; \overline{\mathcal{C}}_{j-1}) - \Delta\right) \tag{19}$$

$$= \frac{1}{k}\sum_{j=1}^{k}\hat{\sigma}_j^{-1}\left((\bar{D}_j(\mathcal{C}_j; \overline{\mathcal{C}}_{j-1}) - \mathbb{E}[\bar{D}_j(\mathcal{C}_j; \hat{d}_{j-1}^{opt}, \mu, \theta, p)|\overline{\mathcal{C}}_{j-1}]) + (\mathbb{E}[\bar{D}_j(\mathcal{C}_j; \hat{d}_{j-1}^{opt}, \mu, \theta, p)|\overline{\mathcal{C}}_{j-1}] - \Delta)\right) \tag{20}$$

$$= \frac{1}{k}\sum_{j=1}^{k}\hat{\sigma}_j^{-1}\left(\bar{D}_j(\mathcal{C}_j; \overline{\mathcal{C}}_{j-1}) - \mathbb{E}[\bar{D}_j(\mathcal{C}_j; \hat{d}_{j-1}^{opt}, \mu, \theta, p)|\overline{\mathcal{C}}_{j-1}]\right) + o_p(k^{-1/2}) \tag{21}$$

$$= \frac{1}{k}\sum_{j=1}^{k}\hat{\sigma}_j^{-1}\left(\bar{D}_j(\mathcal{C}_j; \overline{\mathcal{C}}_{j-1}) - \mathbb{E}[\bar{D}_j(\mathcal{C}_j; \overline{\mathcal{C}}_{j-1})|\mathcal{F}_{j-1}]\right) + o_p(k^{-1/2}). \tag{22}$$

Above (21) follows by condition (C5) and (22) follows by condition (C4). For $j = 1, 2, \cdots, k$, let

$$M_{k,j} = \frac{1}{\sqrt{k}} \cdot \frac{\bar{D}_j(\mathcal{C}_j; \overline{\mathcal{C}}_{j-1}) - \mathbb{E}[\bar{D}_j(\mathcal{C}_j; \overline{\mathcal{C}}_{j-1})|\mathcal{F}_{j-1}]}{\hat{\sigma}_j}. \tag{23}$$

It is obvious that for each $k$, $M_{k,j}$, $1 \le j \le k$, is a martingale with respect to the filtration $\mathcal{F}_j$. In particular, for all $j \ge 1$, $\mathbb{E}[M_{k,j}|\mathcal{F}_{j-1}] = 0$ and $\sum_{j=1}^{k}\mathbb{E}[M_{k,j}^2|\mathcal{F}_{j-1}] = \frac{1}{k}\sum_{i=1}^{k}\frac{\sigma_j^2}{\hat{\sigma}_j^2} \xrightarrow{p} 1$ as $k \to \infty$ by (C3). The conditional Lindeberg condition holds in (C2), so the *martingale central limit theory* for triangular arrays gives

$$\sum_{j=1}^{k} M_{k,j} \xrightarrow{d} N(0,1). \tag{24}$$

Plugging it back into (22), we can get

$$\frac{1}{\sqrt{k}}\left(\sum_{j=1}^{k}\hat{\sigma}_j^{-1}\right)\left(\hat{\Delta}_k - \Delta\right) \xrightarrow{d} N(0,1). \tag{25}$$

### A.1.2 PROOF OF PROPOSITION 1

We first simplify the formula of $\lambda_k$ to:

$$\lambda_k = \frac{\psi_{\left(\frac{1}{\sqrt{k}}(\sum_{j=1}^{k}\hat{\sigma}_j^{-1})\Delta, 1\right)}(R_k)}{\psi_{(0, 1)}(R_k)} \tag{26}$$

$$= \exp\left\{\frac{1}{\sqrt{k}}\sum_{j=1}^{k}\hat{\sigma}_j^{-1} \cdot \Delta \cdot R_k - \frac{1}{2k}(\sum_{j=1}^{k}\hat{\sigma}_j^{-1})^2 \cdot \Delta^2\right\} \tag{27}$$

$$= \exp\left\{\frac{1}{k}\sum_{j=1}^{k}\hat{\sigma}_j^{-1} \cdot \Delta \cdot \sum_{j=1}^{k}(\hat{\sigma}_j^{-1}\bar{D}_j) - \frac{1}{2k}(\sum_{j=1}^{k}\hat{\sigma}_j^{-1})^2 \cdot \Delta^2\right\}, \tag{28}$$

where we denote $\bar{D}_j(\mathcal{C}_j; \overline{\mathcal{C}}_{j-1})$ with $\bar{D}_j$ for simplicity. Let

$$\hat{\Delta}_k := \frac{\sum_{j=1}^{k}\hat{\sigma}_j^{-1}\bar{D}_j}{\sum_{j=1}^{k}\hat{\sigma}_j^{-1}}. \tag{29}$$

and remember that Theorem 3.1 gives

$$\frac{1}{\sqrt{k}} \left( \sum_{j=1}^{k} \hat{\sigma}_j^{-1} \right) \left( \hat{\Delta}_k - \Delta \right) \xrightarrow{d} N(0,1), \tag{30}$$

where $\hat{\sigma}_j$ is estimated from the first $j$ batches $\overline{\mathcal{C}}_{j-1}$, $j = 1, 2, \cdots, k$. Since the true value difference $\Delta$ is not very large in practice, we assume local alternative $\Delta = O_p(k^{-1/2})$ here as in Theorem 3.1. Then,

$$\mathbb{E}_{H_0}[\lambda_{k+1}|\mathcal{F}_k] \tag{31}$$

$$= \mathbb{E}_{H_0} \left\{ \exp \left[ \frac{1}{k+1} \sum_{j=1}^{k+1} \hat{\sigma}_j^{-1} \cdot \Delta \cdot \sum_{j=1}^{k+1} (\hat{\sigma}_j^{-1} \bar{D}_j) - \frac{1}{2(k+1)} (\sum_{j=1}^{k+1} \hat{\sigma}_j^{-1})^2 \cdot \Delta^2 \right] \middle| \mathcal{F}_k \right\} \tag{32}$$

$$\overset{\text{Delta Method}}{\approx} \exp \left\{ \mathbb{E}_{H_0} \left[ \frac{1}{k+1} \sum_{j=1}^{k+1} \hat{\sigma}_j^{-1} \cdot \Delta \cdot \sum_{j=1}^{k+1} (\hat{\sigma}_j^{-1} \bar{D}_j) - \frac{1}{2(k+1)} (\sum_{j=1}^{k+1} \hat{\sigma}_j^{-1})^2 \cdot \Delta^2 \middle| \mathcal{F}_k \right] \right\} \tag{33}$$

$$= \exp \left\{ \frac{1}{k+1} \sum_{j=1}^{k+1} \hat{\sigma}_j^{-1} \cdot \Delta \cdot \left( \sum_{j=1}^{k} (\hat{\sigma}_j^{-1} \bar{D}_j) + \hat{\sigma}_{k+1}^{-1} \cdot \underbrace{\mathbb{E}_{H_0}[\bar{D}_{k+1}|\mathcal{F}_k]}_{0} \right) - \frac{1}{2(k+1)} (\sum_{j=1}^{k+1} \hat{\sigma}_j^{-1})^2 \cdot \Delta^2 \right\} \tag{34}$$

$$= \exp \left\{ \frac{1}{k+1} \sum_{j=1}^{k+1} \hat{\sigma}_j^{-1} \cdot \Delta \cdot \sum_{j=1}^{k} (\hat{\sigma}_j^{-1} \bar{D}_j) - \frac{1}{2(k+1)} (\sum_{j=1}^{k+1} \hat{\sigma}_j^{-1})^2 \cdot \Delta^2 \right\} \tag{35}$$

$$= \exp \left\{ \frac{1}{k} \sum_{j=1}^{k} \hat{\sigma}_j^{-1} \cdot \Delta \cdot \sum_{j=1}^{k} (\hat{\sigma}_j^{-1} \bar{D}_j) + \left( \frac{1}{k+1} \sum_{j=1}^{k+1} \hat{\sigma}_j^{-1} - \frac{1}{k} \sum_{j=1}^{k} \hat{\sigma}_j^{-1} \right) \cdot \Delta \cdot \sum_{j=1}^{k} (\hat{\sigma}_j^{-1} \bar{D}_j) \right.$$

$$\left. - \frac{1}{2k} (\sum_{j=1}^{k} \hat{\sigma}_j^{-1})^2 \cdot \Delta^2 - \left( \frac{1}{2(k+1)} (\sum_{j=1}^{k+1} \hat{\sigma}_j^{-1})^2 - \frac{1}{2k} (\sum_{j=1}^{k} \hat{\sigma}_j^{-1})^2 \right) \Delta^2 \right\} \tag{36}$$

$$= \lambda_k \cdot \exp \left\{ \left( \frac{1}{k+1} \sum_{j=1}^{k+1} \hat{\sigma}_j^{-1} - \frac{1}{k} \sum_{j=1}^{k} \hat{\sigma}_j^{-1} \right) \cdot \Delta \cdot \sum_{j=1}^{k} (\hat{\sigma}_j^{-1} \bar{D}_j) - \left( \frac{1}{2(k+1)} (\sum_{j=1}^{k+1} \hat{\sigma}_j^{-1})^2 - \frac{1}{2k} (\sum_{j=1}^{k} \hat{\sigma}_j^{-1})^2 \right) \Delta^2 \right\} \tag{37}$$

$$\overset{(29)}{=} \lambda_k \cdot \exp \left\{ \left( \frac{1}{k+1} \sum_{j=1}^{k+1} \hat{\sigma}_j^{-1} - \frac{1}{k} \sum_{j=1}^{k} \hat{\sigma}_j^{-1} \right) \cdot \Delta \cdot \sum_{j=1}^{k} \hat{\sigma}_j^{-1} \cdot \hat{\Delta}_k - \left( \frac{1}{2(k+1)} (\sum_{j=1}^{k+1} \hat{\sigma}_j^{-1})^2 - \frac{1}{2k} (\sum_{j=1}^{k} \hat{\sigma}_j^{-1})^2 \right) \Delta^2 \right\} \tag{38}$$

$$= \lambda_k \cdot \exp \left\{ \underbrace{\left( \frac{1}{k+1} \sum_{j=1}^{k+1} \hat{\sigma}_j^{-1} - \frac{1}{k} \sum_{j=1}^{k} \hat{\sigma}_j^{-1} \right)}_{O_p(k^{-1})} \times \underbrace{\sum_{j=1}^{k} \hat{\sigma}_j^{-1} \hat{\Delta}_k}_{O_p(k^{1/2}) \text{ by } (30)} \times \Delta \right.$$

$$\left. - \underbrace{\left( \frac{1}{2(k+1)} (\sum_{j=1}^{k+1} \hat{\sigma}_j^{-1})^2 - \frac{1}{2k} (\sum_{j=1}^{k} \hat{\sigma}_j^{-1})^2 \right)}_{O_p(1)} \times \Delta^2 \right\} \tag{39}$$

$$= \lambda_k \cdot \exp\{o_p(1)\} \tag{40}$$

Table 3: Estimated type I error or power for SUBTLE and SST with batch size 40

| Model | I | | II | | III | | IV | | V | |
|---|---|---|---|---|---|---|---|---|---|---|
| $c$ | SUBTLE | SST | SUBTLE | SST | SUBTLE | SST | SUBTLE | SST | SUBTLE | SST |
| -1 | 0.003 | 0.662 | 0.000 | 0.588 | 0.000 | 0.219 | 0.000 | 0.368 | 0.002 | 0.006 |
| 0 | 0.012 | 0.126 | 0.002 | 0.023 | 0.003 | 0.077 | 0.002 | 0.023 | 0.006 | 0.034 |
| 0.6 | 0.297 | 0.552 | 0.465 | 0.549 | 0.326 | 0.397 | 0.414 | 0.451 | 0.585 | 0.216 |
| 0.8 | 0.633 | 0.837 | 0.868 | 0.896 | 0.680 | 0.703 | 0.826 | 0.816 | 0.931 | 0.373 |
| 1 | 0.901 | 0.969 | 0.993 | 0.995 | 0.947 | 0.933 | 0.985 | 0.978 | 0.999 | 0.715 |

Table 4: Estimated type I error and power for SUBTLE and SST with varying mixture density variance

| $\tau^2$ | 0.0001 | 0.001 | 0.01 | 0.1 | 1 | 10 |
|---|---|---|---|---|---|---|
| Type I error | 0.003 | 0.024 | 0.021 | 0.016 | 0.005 | 0.002 |
| Power | 0.887 | 0.976 | 0.956 | 0.932 | 0.892 | 0.825 |

## A.2 ADDITIONAL RESULTS

### A.2.1 HYPERPARAMETERS

There are three hyperparameters in our algorithm: batch size $m$, variance of mixture density $\tau^2$, and failure time $M$. We did not tune these hyperparameters in our experiments, but used the same value for SST and SUBTLE. In the following, we will expound the effects of these hyperparameters on the performance of our tests and provide additional simulation results.

Apart from batch size 20 in Section 4.1, we also conduct experiments with batch size 40 under the same setting. The results are shown in Table 3. It seems that there is considerable robustness in choosing batch size.

In theory the choice of mixture density variance $\tau^2$ will not have any effect on the type I error control. Johari et al. (2015) proved that an optimal $\tau^2$ in terms of stopping time is the prior variance times a correction for truncating. It is the reason that we suggest using historical data to estimate the variance of value difference $\Delta$. Besides, we conduct simulations with varying $\tau^2$. The data is generated from Model I in Table 1 with $c = 0$ or $c = 1$. When $c = 0$ we estimate the type I error, while when $c = 1$ we estimate the power. The results in Table 4 show that the type I error is always controlled below significance level 0.05 and the power has considerable robustness to the choice of $\tau^2$.

As we mentioned in future work, how to choose the optimal failure time $M$ is still a problem. The larger the failure time, the higher power we have to detect the difference since we collect more samples. However, large failure time also means long waiting time and high opportunity cost. Thus, there is a trade off between waiting time and power.

### A.2.2 OPTIMAL TREATMENT RULE FOR YAHOO DATA

Figure 3 gives the decision tree of the estimated optimal treatment rule. Each left branch contains the subpopulation whose covariates satisfy the conditions on its parent node. The classification 0/1 on each leaf node indicates the optimal treatment rule for corresponding subpopulation, and the two values separated by slash gives the number of users who "truly" (estimated by random forest) benefit from control and treatment.

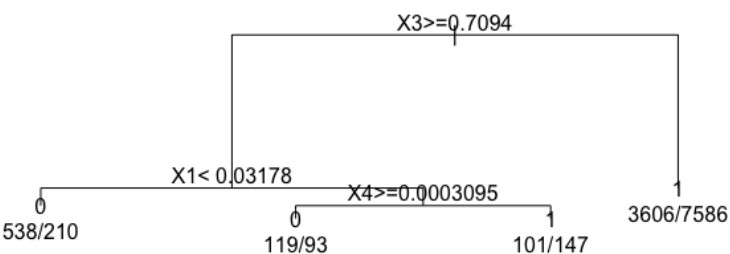

Figure 3: The decision tree of optimal treatment rule for Yahoo data

