# OpenReview forum: "Online Testing of Subgroup Treatment Effects Based on Value Difference"
_ICLR.cc/2021/Conference — Reject_

### Official Review · AnonReviewer4 · 2020-10-17
**Official Blind Review #4**

**Rating:** 7
**Confidence:** 3

**Review:**

*Summary*

This paper proposes a new algorithm (SUBTLE) to conduct online A/B testing that 1) allows for continuous monitoring, and 2) detects subgroups with enhanced treatment effect (if such subgroups exist). The authors formalize the problem into a clean hypothesis testing problem (9) that tests if the value gap between the optimal policy and the all-control policy is 0, propose the algorithm SUBTLE, and prove that it is able to control type I error at any time. Experiment results compare SUBTLE with the existing baseline SST, and show that SUBTLE is able to control type I error (while SST fails when the treatment has clear negative impact) and achieves competing detection power.

*Overall Assessment*

Overall I vote for acceptance.  This paper focuses on an important real-world problem that many ICLR readers care about, and is easy to follow in general.

*Pros*
- The problem considered in this paper looks real and relevant to me. I like the idea of "testing the existence of a beneficial subgroup", which should be more useful in practice than "testing whether there's difference between treated and untreated" as SST does. The testing hypotheses (9) look clean and reasonable to me.
- The proposed algorithm (SUBTLE) adapts mSPRT to address the current problem, which is easy to understand and achieves provable type I error guarantees. It also enjoys relatively good performance in both simulated and real data experiments.
- The paper is clearly written and easy to follow.

*Cons and Questions*
- SUBTLE performs extremely well in model V (the high-dimensional model), achieving lower type I error and higher power than in model I-IV.  I'm kind of doubtful about this result, since model V is essentially like model I-IV with more noise covariates, and as shown in section 4.2 adding more noise covariates lowers the estimated power. Is there any good explanation for this? Is it because the treatment effect structure in model V makes random forest a more favorable method? If so, how does SUBTLE perform when we have different $\theta$ structures (like linear)?
- SUBTLE looks relatively more conservative than SST when $c$ is small.  In many real-world applications, treatment effect $\theta(X)$ is often smaller in magnitude than $\mu(X)$, which is like small a $c$ in experiments. I am curious about how will the results look like when $c$ is smaller than $0.6$.

---

> ### Author Response · Authors · 2020-11-17
> **Thanks for your comments. The replies to each comment are as below.**
>
> Thanks for your comments. The replies to each comment are as below:
>
> 1. The power of SUBTLE largely depends on the true value difference $\Delta$, which is determined by the intensity of the value difference (i.e. $c$ in $\theta(X)$), and the size of the beneficial subgroup. Since we compare five models using the same $c$, the difference in power is caused by the size of the beneficial subgroup. For example when $c>0$ the beneficial subgroup for model II \& III is $1\\{(X_2 > 0) | (X_5 < -0.5)\\}$, which accounts for 65.5\% of the population since
> $$
> P((X_2 > 0)|( X_5 < -0.5)) = P(X_2 > 0) + P(X_5 < -0.5) - P((X_2>0) \\& (X_5 < -0.5)) =  0.655.
> $$ Similarly, the beneficial subgroup $1\\{X_1 + 2X_3 >0\\}$ for model I \& IV accounts for 67\% of the population while the beneficial subgroup $1\{(X_{14} > -0.1) \\&(X_{20} =1)\}$ for model V accounts for 72\% (=0.8*0.9). The large beneficial subgroup of model V can explain its good performance over other models even with more noise variables. Besides, the accuracy of all the estimates, which may be impacted by different structures of $\mu$ and $\theta$, the covariates distributions, and number of noise covaraites, also makes a difference in the performance (type I error and power) of SUBTLE.
>
> 2. First of all, it should be noted that SST is not a valid test for our considered hypothesis of subgroup existence (i.e. it can't control the type I error). This is due to various reasons but one main reason is that the assumed parametric model in SST is misspecified even under the null when $c = 0$. So there is no much point to compare the type I errors of SUBTLE and SST. Secondly, the conservativeness (i.e. the type I error is much less than 0.05) of SUBTLE is by the nature of sequential testing that allows continuous monitoring. The validity of such continuously monitored tests, like SUBTLE, mSPRT and SST, relies on the application of Markov's inequality and optimal stopping theorem for martingales, which controls type I error under the nominal level as shown in the paper. But in general, the Markov's inequality is conservative and the equality can't be achieved. That is why these tests have conservative type I errors but this is the price we have to pay with the continuous testing. Finally in industry, launching new products or features costs money and labors, so we are only interested in detectable effect size that is large enough to have these cost paid back. That is why we chose $c$ from 0.6 to 1 to illustrate the power of our proposed test. To be complete, following your suggestion, we also consider settings with $c = 0.3$ and the power are 0.054, 0.063, 0.052, 0.053, 0.066 for Model I-V.

---

### Official Review · AnonReviewer1 · 2020-10-22
**Many design choices need to be explained. The paper does not seem to fall in the scope of ICLR.**

**Rating:** 3
**Confidence:** 4

**Review:**

Summary:
This paper provides a statistical test that checks whether a subgroup would benefit from a certain intervention, even if on average the entire population does not benefit from that intervention. This test is developed for the online setting where we have a stream of data coming in and we want to terminate the trial (here, A/B testing) as soon as possible in order to limit the possible adverse effects that the trial might have.

Pros:
- The paper is well-written and easy to read / understand.
- According to the paper’s literature review, this work seems to be original.

Cons:
- There are many design choices that the authors do not discuss the reasons behind making those choices; an example is: employing the augmented inverse probability weighted (AIPW) estimators.
- It is unclear why $\Delta$ should be defined as that in Eq. (11).
- There is no step-by-step discussion on how the test statistic is derived as that in Eq. (12) or (15). By the way, are these two equations equivalent? If so, how?
- Most importantly, I think the scope of this paper does not match with that of the ICLR conference, as there is no representation learning aspect in this work. Perhaps AISTATS, or even IJCAI or AAAI would be a better fit to publish this paper.

Minor comments:
- Sec. 3.1: $g(\mu)=\log(\frac{\mu}{1-\mu})$ is undefined for a binary $\mu$.
- Item (4) in Algorithm 1: $\overline{D}_k$ is a scalar and has no standard deviation.

---

> ### Author Response · Authors · 2020-11-17
> **Thanks for your comments. However, we disagree with some reviews.**
>
> Thanks for your comments. Replies to each comment are as below:
>
> 1. The AIPW estimator $\hat{V}(d)$ for a value function $V(d)$ of a treatment decision rule $d$ is in the following form:
> $$
> \frac{1}{n} \sum_{i=1}^n  \\{\frac{Y_i 1_{A_i=d}}{p_{A_i}(X_i)}- (\frac{1_{A_i=d}}{p_{A_i}(X_i)} - 1) \cdot E[Y_i|X_i, A_i=d]\\}
> $$
> where $p_{A}(X)=p(X)*A + (1-p(X)) * (1-A)$ and $p(X)$ is the propensity score. It is a widely-used unbiased estimator, i.e., $E_{(Y,A,X)}[\hat{V}(d)] = V(d)$. Moreover, the most important property of AIPW estimator is the double robustness, that is, the estimator remains consistent if either the estimator of $E[Y_i|X_i, A_i=d]$ or the estimator of the propensity score $p(X)$ is consistent, which gives much flexibility. We have added some discussions on AIPW estimator in the paper. Please let us know what other design choices we should discuss.
>
> 2. We have mentioned in Section 3.2 that $D (O_i; \mu, \theta, p)$ (10), which equals $\hat{V}(d^{opt}) - \hat{V}(0)$, is an unbiased estimator for $\Delta$. However,  $\mu, \theta, p$ in $D (O_i; \mu, \theta, p)$ are usually unknown, so we replace it with $D (O_i; \hat{\mu_{k-1}}, \hat{\theta_{k-1}}, \hat{p_{k-1}})$, where $\mu, \theta, p$ are estimated. Then $\bar{D}_j(C_j; \bar{C}_{j-1})$ (13) as an average of
>
> $D (O_i; \hat{\mu_{k-1}}, \hat{\theta_{k-1}}, \hat{p_{k-1}})$ over one batch is also an estimator for $\Delta$, and so is the weighted average of $\bar{D}_j(C_j; \bar{C}_{j-1})$, which is exactly $\hat{\Delta}_k $ (11).
>
> 3. We actually gave these discussion on page 4 and 5. $R_k$ (14) is a multiplier of $\hat{\Delta}_k$ (11), where $\hat{\Delta}_k$ is an asymptotic unbiased estimator for the testing parameter $\Delta$ (9). We also showed in section 3.3 that $R_k$ (14) has an asymptotic normal distribution with same variance but different means under null and local alternatives, so that our test statistics $\Lambda_k^{\pi}$ (15) is constructed as a mixture asymptotic probability ratios of $R_k$. This idea comes from mSPRT (2) and by this construction we are able to control the type I error with same decision rule (3) as mSPRT. After some calculations, it can be shown that the equation (12) is equivalent to (15) with $\pi(\Delta)=\frac{2}{\sqrt{2\pi\tau^2}}\cdot \exp{\\{-\frac{\Delta^2}{2\tau^2}\\}}\cdot 1(\Delta > 0)$.
>
> 4. We don't agree with the review on this point. To be specific, our paper falls into the scope of supervised representation learning, which is a topic that is of interest to ICLR. We consider a situation where each user is (random or observational) assigned to one of the two variants (control and treatment) and has an observed outcome. Based on each user's observed outcome, assigned variant and covaraites, we hope to classify the population into a sub-population which would benefit from the treatment and a sub-population which would not. And for a new observation, we hope to predict which sub-population it falls into based on its covariates and provide the optimal variant to it. Our proposed algorithm automatically detects from the streaming data whether a subgroup which benefits from a particular treatment exists or not based on a testing procedure, and then identifies this subgroup using random forest or decision tree if such subgroup exists. It is easy to implement and has wide application in A/B testing.
>
> 5. $g(\mu)=\log \frac{\mu}{1-\mu}$ is well-defined. The $\mu$ is the mean of $Y$ where $Y$ is binary data, so $\mu$ is some value greater than 0 and less than 1, and $g(\mu)$ maps $\mu$ to ($-\infty$, $\infty$).
>
> 6. $\bar{D}_j(C_j; \bar{C}_{j-1})$ (13) is not a constant but a random variable, since it contains data $O_i$.

---

> > ### Comment · AnonReviewer1 · 2020-11-24
> > **Thank you for your responses; however, my main comment stands.**
> >
> > I would like to thank the authors for their rebuttal.
> > However, I still think that ICLR is not the right venue for publishing this paper, as there is no **representation learning** component in the work.

---

### Official Review · AnonReviewer2 · 2020-10-26
**Interesting paper with room for improvement**

**Rating:** 5
**Confidence:** 4

**Review:**

In this paper, the authors propose SUBTLE, which performs sequential A/B test with heterogeneous treatment effect. Compared to prior work, SUBTLE does not require specification of the parametric from of the treatment effect on some covariates.

I believe the paper is original and could be useful for practitioners. The paper could be improved in clarity. See my detailed comments below. I believe most of the my comments could be easily addressed, and a few of them may require some more substantial work.

On page 2, the authors state "it generally gives a significant decrease in the required sample size compared to the fixed-horizon test..." I don't think this statement is true. The confidence level required to terminate the experiment early is usually required to be higher than the nominal alpha level. Furthermore, because we peek the experiment multiple times, the required confidence level at the final analysis point is also higher than the nominal alpha level. Thus, the required sample size for sequential testing is usually higher than that with classical fixed-horizon test.

On page 3, is there any constraint on \mathcal{X}_0? It seems from (8) that it could theoretically be possible that \mathcal{X}_0 is a set of disjoint points. In this case, even if the null hypothesis is rejected, we cannot practically launch the treatment. A more practical alternative hypothesis is either to constrain that \mathcal{X}_0 is convex, or that for all x in \mathcal{X}_0, theta(x) > 0 AND theta(x) >= 0 for all x (non-inferiority).

On page 4, I am confused by the big X notation. Is the probability/expectation on X taken with respect to the distribution of an individual observation X_i, a subgroup of observations \mathcal{X}_0, or the distribution of all the observations \mathcal{X}? On a related question, is Delta an observation-dependent value, a subgroup-dependent value, or a fixed value common for the population? Based on the formulation of (9), I suppose Delta is a population level value, but this will imply that V(d^opt) and V(0) are the same across subgroups with heterogeneous treatment effect. Is this reasonable? Am I missing anything?

On page 7, if I read Table 2 correctly, the rows with c=0/c=-1 show the type-I error rates of SUBTLE and SST. Why is SUBTLE so conservative given that alpha = 0.05 and the Type-I error rate are all greatly below 0.05?

On page 7, it will be good to include another simulated experiment to use random forest to estimate \mathcal{X}_0. The authors did this on the real (private) dataset on page 8, but it would be good to do the same on a dataset where it is publicly available and the ground truth is known.

On page 10, what is the meaning of C4? How likely are C4 and C5 to hold? It would be good for authors to give an intuitive explanation, and, given that these two conditions are not standard in literature, run some numerical studies to directly validate these two conditions. Also, since the conditions are important, I suggest the writer to move the conditions to the main paper from appendix.

---

> ### Author Response · Authors · 2020-11-17
> **Thanks for your comments. Replies to each comment are as below.**
>
> Thanks for your comments. Replies to each comment are as below:
>
> 1. We have revised this statement as "It generally gives a significant decrease in the required sample size compared to the fixed-horizon test with the same type I error and type II error control" and this has been widely studied in the literature. For example, Wald (1945) proved that the SPRT minimizes the expected sample size under both null and alternative hypotheses among all the tests with the same type I error and type II error control. We also showed in Section 4.3 that SUBTLE generally requires smaller sample size to stop the experiment compared with its corresponding fixed-horizon test with the same power and significance level.
>
> 2. We require the subgroup set $\mathcal{X}_0$ needs to have a positive probability, that is $P(X \in \mathcal{X}_0) >0$. A set of disjoint points has measure zero and does not satisfy this condition. We have added this condition in the paper.
>
> 3. The probability/expectation on $X$ is taken with respect to the distribution of the whole population $\mathcal{X}$. We assume that the triplet $O_i = \\{Y_i, A_i, X_i\\}$ are independent and identically distributed across $i$. The expectation in the paper, unless otherwise specified, is taken with respect to the joint distribution of the observed data $(Y, A, X)$. For example, the value function $V(d)$ is the expectation of $Y^*(d(X))$ with respect to the joint distribution of the potential outcome and covariates $(Y^*(a), X)$, and can be shown to be equivalent to $E_{X} \\{ E_{Y|A,X}[Y|A=d(X), X] \\}$ under two standard causal inference assumptions: (i) no unmeasured confounders, i.e. the potential outcome $Y^*(a)$ is independent of the treatment $A$ given covariates $X$; (ii) consistency, i.e. the potential outcome $Y^*(a)$ of a treatment is equal to the observed outcome $Y$ under that treatment.
> The value difference $\Delta$ is defined on the whole population and it equals  $E_{X} \\{ E_{Y|A,X}[Y|A=d^{opt}(X), X] \\}-E_{X} \\{ E_{Y|A,X}[Y|A=0, X] \\}$. When there exists a beneficial subgroup in the population, $V(d^{opt})$ is greater than $V(0)$ and $\Delta$ is positive. Otherwise, $\Delta = 0$ under the null. This is the reason why we can test the considered hypothesis of subgroup existence based on a consistent estimator of $\Delta$. We have added the above discussions in the paper.
>
> 4. The conservativeness (i.e. the type I error is much less than 0.05) of SUBTLE is by the nature of sequential testing that allows continuous monitoring. The validity of such continuously monitored tests, like SUBTLE, mSPRT and SST, relies on the application of Markov's inequality and optimal stopping theorem for martingales, which controls type I error under the nominal level as shown in the paper. But in general, the Markov's inequality is conservative and the equality can't be achieved. That is why these tests have conservative type I errors but this is the price we have to pay with the continuous testing.
>
> 5. For each model in Section 4.1, we sample $M=2300$ observations and identify the beneficial subgroup $1\\{\theta(X) > 0\\}$ by estimating $\theta(X)$ with random forest. We then build a classification tree on the same data with random forest estimator $1\\{\hat{\theta}(X) > 0\\}$ as the true labels. The obtained structured optimal treatment rules are shown in https://www.dropbox.com/sh/q6ilkectztg3m8y/AACPZqAACj9RE5YwwKT4Nh7la?dl=0.
>
> 6. We move these conditions to the main paper. The conditions (C4) and (C5) actually are standard conditions used in the literature for studying the properties of the doubly robust estimators. For example, Luedtke et al (2016) used similar conditions and discussed their appropriateness.  Both (C4) and (C5) relies on the convergence rate of the estimator of $\mu, \theta, p$. The type I error in our simulation is well-controlled, which demonstrates that the convergence of $\hat{\theta}, \hat{\mu}, \hat{p}$ to $\theta, \mu, p$ is fast enough to make (C4) and (C5) satisfied. The intuitions behind (C4) and (C5) are as follows.
> In (C4), we look at the difference in the expectation of the average of $D(O_i; \hat{\mu}, \hat{\theta}, \hat{p})$ and the average of $D(O_i; \hat{d}^{opt}, \mu, \theta, p)$ over one batch. The difference between these two terms is that in $D(O_i; \hat{d}^{opt}, \mu, \theta, p)$ only the optimal treatment rule $d^{opt}=1\\{\theta(X) > 0\\}$ is estimated while in $D(O_i; \hat{\mu}, \hat{\theta}, \hat{p})$ all the $\mu, \theta, p$ are estimated. Therefore, if $\\|\hat{f}(X) - f(X)\\|_{2,P}  \xrightarrow{P} 0$ under certain rate for all $f(\cdot) \in \\{\mu(\cdot), \theta(\cdot), p(\cdot)\\}$, we are able to get (C4).
>
> For (C5), since $\Delta=E_{(Y,A,X)}[D(O_{i}; \mu, \theta, p)]= E_{(Y,A,X)}[\bar{D}_j(\mathcal{C}_j; \mu, \theta, p)]$,
> we only require that the estimated optimal treatment rule $\hat{d}^{opt}(X)=1\\{\hat{\theta}(X) > 0\\}$ converges to the true $d^{opt}(X) = 1\\{\theta(X)>0\\}$ at a certain rate.

---

### Official Review · AnonReviewer3 · 2020-10-29
**Online Testing of Subgroup Treatment Effects Based on Value Difference**

**Rating:** 7
**Confidence:** 5

**Review:**

This paper is to propose a sequential test for subgroup treatment effects based on value difference, named SUBTLE,
to address these two problems including  a fixed-horizon framework and  identifying a subgroup with a beneficial treatment effect。
Although there are several interesting results, this paper is full of many typos and small errors. Here are some detailed comments.

1. Please not use (1)-(?) to itemize different things, since you also use them for equations.

2. The estimator in (10) still suffers from the instability of the estimated propensity score.

---

> ### Author Response · Authors · 2020-11-17
> **Thanks for your comments. Replies to each comment are as below.**
>
> Thanks for your comments. Replies to each comment are as below:
>
> 1. Thanks for the suggestions. We have revised these notation in the paper to make them clearer.
>
> 2. We have added some standard causal inference assumptions in the paper, including the positivity assumption, i.e. $\mathbb{P}(A=a|X=x) >0$ for $a=0,1$ and all $x \in \mathcal{X}$ such that $\mathbb{P}(X=x)>0$. When the sample size is large enough, the estimated propensity score is also bounded from 0 and 1 and will not suffer the instability issue as we have observed in simulations and real data application.

---

### Decision · Program_Chairs · 2021-01-07
**Final Decision**

**Decision:**

Reject

**Comment:**

In this paper, the authors propose a test for subgroup treatment effects in settings where data is obtained online, via a method they call SUBTLE.  The authors adopt a semi-parametric (generalized linear model) approach to modeling nuisance functions.  The authors derive the form of the distribution of their test statistic in (12), which is based on asymptotic normality the influence function based estimator.

The authors evaluate their methods via simulation studies, and on a dataset of user clicks from Yahoo!

The opinion of the reviewers was somewhat split on this paper.  One reviewer felt the paper was out of scope for ICLR, although this did not influence the overall evaluation of the paper -- since ICLR's scope has now broadened and solicits work on all areas of machine learning and related areas of data science (as the ICLR website makes clear).

However, reviewers raised a number of concerns about the paper (in particular, see reviewer 2) that on balance did not persuade them that the paper is ready for publication in the current state.